# Improving Well-Being in Young Adults: A Social Marketing Proof-of-Concept

**DOI:** 10.3390/ijerph19095248

**Published:** 2022-04-26

**Authors:** Yannick van Hierden, Sharyn Rundle-Thiele, Timo Dietrich

**Affiliations:** Social Marketing @ Griffith, Griffith Business School, Griffith University, Nathan, QLD 4111, Australia; s.rundle-thiele@griffith.edu.au (S.R.-T.); t.dietrich@griffith.edu.au (T.D.)

**Keywords:** well-being, health promotion, behavior change, social marketing, evaluation, theory, intervention, social cognitive theory

## Abstract

Approximately 1 in 5 Australians experience a mental disorder every year, costing the Australian economy $56.7 billion per year; therefore, prevention and early intervention are urgently needed. This study reports the evaluation results of a social marketing pilot program that aimed to improve the well-being of young adults. The *Elevate Self Growth* program aimed to help participants perform various well-being behaviors, including screen time reduction, quality leisure activities, physical activity, physical relaxation, meditation and improved sleep habits. A multi-method evaluation was undertaken to assess *Elevate Self Growth* for the 19 program participants who paid to participate in the proof-of-concept program. Social Cognitive Theory was used in the program design and guided the evaluation. A descriptive assessment was performed to examine the proof-of-concept program. Considerations were given to participants’ levels of program progress, performance of well-being behaviors, improvements in well-being, and program user experience. Participants who had made progress in the proof-of-concept program indicated improved knowledge, skills, environmental support and well-being in line with intended program outcomes. Program participants recommended improvements to achieve additional progress in the program, which is strongly correlated with outcome changes observed. These improvements are recommended for the proof-of-concept well-being program prior to moving to a full randomized control trial. This paper presents the initial data arising from the first market offerings of a theoretically mapped proof-of-concept and reports insights that suggest promise for approaches that apply Social Cognitive Theory in well-being program design and implementation.

## 1. Introduction

For decades, mental disorders have been leading causes of the health-related burden world-wide [1,2]. Approximately 1 in 5 Australians experience mental disorders, costing the Australian economy $56.7 billion per year [3,4]. Prior to the COVID-19 pandemic, between 1980 and 2004, the prevalence of anxiety and depression in young people in the United Kingdom increased by 70% [5]. Recent studies have demonstrated a further significant increase in anxiety and psychological distress as a result of the COVID-19 pandemic [2,6,7]. The greater increase in anxiety and depression will add substantially to an already significant and costly problem. Given many psychological problems (e.g., stress, anxiety and depression) are preventable [8,9], enhancing well-being for young adults through prevention is urgently needed.

Scholars and experts are asking for new health interventions to support individuals’ mental well-being [10]. Existing well-being interventions generally target health behaviors, including nutrition, physical activity, deep breathing exercises, relaxation exercises, mindfulness, meditation, gratitude practices (e.g., letter writing, journaling, three good things) and goal setting [11,12,13,14,15,16,17,18]. In a previous review of 13 health interventions, eight claimed that their intervention was informed by theory; however, only four interventions specified the use of theory-based behavior-change methods, and only one intervention specified exactly how such behavior-change methods were mapped to the theory listed. Further, many health interventions are not based on theory, despite the fact that behavioral and social science disciplines recommend using theory in the design and implementation thereof. Evaluations of behavior-change interventions have demonstrated improved effects when theory use was reported [19,20,21]. Therefore, mapping theory to a behavior-change intervention is expected to improve effectiveness [21,22,23], yet theories are still underused in behavior-change intervention development, and application of social marketing to enhance well-being is underreported [24,25,26]. Of the interventions that claim to be theory-informed, many have failed to report how theory was mapped and make it difficult to assess the effectiveness of the theory base. Further, health interventions require the involvement of program participants in program design, implementation and evaluation. However, interventions are often developed in a ‘top-down’ manner without the involvement of program participants [26]. Considering the need for new co-created, theory-based health interventions, a well-being program based on Social Cognitive Theory (SCT) [27] was developed and implemented. The paragraphs hereafter describe how this theory-based well-being intervention was co-created.

Social marketing is distinct from other behavior-change approaches which tell rather than sell. In 2002, Andreasen [28] outlined core characteristics of social marketing, aiming to contrast social marketing from other behavior-change approaches such as public health. More recently, Rundle-Thiele et al. [29] provided an explanation of the strategic process demonstrating how people are placed at the center of program creation, demonstrating when and where core marketing functions were applied in first-time program development. The well-being program was co-created and built prior to engaging people. Co-creation is a critical element of social marketing [29,30]. During co-creation, consideration of *competition* is given by examining how other programs deliver intended outcomes. Ideas gathered are considered by potential program users during participatory design. This intervention was *theoretically informed*, and *formative research* was undertaken prior to developing a program featuring *marketing mix elements* that was sold (*exchange*). The only social marketing element initially described by Andreasen that is missing is *segmentation*.

### 1.1. Co-Creating the Well-Being Program

Co-creation is the first phase in first-time application of social marketing [29]. A participatory design (PD) process was applied to ensure the program was designed with (and not for) young people. To understand what young adults aged 18–35 years preferred in relation to improving well-being, the team undertook a PD study with 57 university students. A full description of how the PD sessions were conducted can be found in research by van Hierden et al. [26]. For the participants, well-being was defined as “the state of being healthy and happy, and experiencing purpose and meaning”. The PD process led to three insights that informed the program design of *Elevate Self Growth*. First, participants preferred to have a focus on positive language (e.g., speaking of mental health rather than mental illness) and increasing positive outcomes rather than decreasing negative outcomes (e.g., increasing resilience rather than reducing anxiety and depression). Second, participants made it clear that they wanted to have opportunities for active learning (e.g., assessments, habit builders, trackers, challenges, tools, etc.). Third, people expressed a need for social interaction and connection through social gatherings, group workshops or online communities. During the PD sessions, intervention components were designed, evaluated and improved and, ultimately, a final proof-of-concept organized into five core program modules emerged.

### 1.2. Applying Theory to the Well-Being Program

The five-step BUILD process was applied to map theory onto the proof-of-concept well-being program [31]. The program sought to improve five areas of well-being: namely, resilience, positive relationships, mindfulness, stress management and purpose and meaning. Informed by the PD research, it was expected that these areas of well-being could be improved by focusing on 10 behaviors: namely, screen time reduction, quality leisure activities, physical activity, physical relaxation, meditation, improved sleep habits, breathwork, meditation, acts of kindness, community and social service and reflective journaling. This evaluation only reports the six well-being behaviors that formed the focus of modules completed by program participants (screen time reduction, quality leisure activities, physical activity, physical relaxation, meditation and improved sleep habits).

As a part of the five-step BUILD process, a theory selection process identified SCT as the most suitable theory to underpin the intervention. The pilot program was developed using a novel theory-mapping process that mapped behavior-change methods and intervention components to SCT constructs [27]. The theory-driven intervention aimed to improve well-being behavior, from a theory-informed standpoint. According to SCT, knowledge, skills and environment are three aspects that need to be considered within an intervention. For example, to enable participants to reduce their screen time (behavior), various intervention components were created to improve knowledge and skills related to screen time reduction, while other intervention components focused on facilitating environmental change to deliver support that enabled participants to reduce their screen time. At least one intervention component was designed to address each individual SCT construct. Given that the intervention aimed to improve 10 behaviors, this process was repeated 10 times, leading to more than 30 intervention components. A full description of how SCT underpinned the intervention design can be found in research by van Hierden et al. [31]. SCT was used to evaluate the intervention, specifically measuring changes in knowledge, skills and environmental support before and after with follow-up timelines ranging from 8 to 12 months. Then, knowledge, skills and environmental support were measured against targeted well-being behaviors: namely, screen time reduction, quality leisure activities, physical activity, physical relaxation, meditation and improved sleep habits.

### 1.3. Implementing the Well-Being Program

The proof-of-concept program was designed at the end of 2019 and was intended to be delivered in person. However, the COVID-19 pandemic restricted in-person gatherings; therefore, the proof-of-concept was built and trialed online. During the trial, 24 people received free access to the program, and they provided feedback to improve material and delivery format. The free trial was followed by the initial market offering for the proof-of-concept *Elevate Self Growth* program, which aimed to engage people who were interested in improving their well-being. The *Elevate Self Growth* program offers videos, assessments, tools and a community to build healthier habits. This study examines data derived from delivery of the proof-of-concept *Elevate Self Growth* program, which occurred over a 12-month period. The primary objective of this study is to evaluate the utility of using SCT in a social marketing program that aims to improve well-being. This article describes the lessons learned from delivery of the proof-of-concept, theory-driven social marketing intervention.

## 2. Materials and Methods

A multi-method evaluation was conducted using self-report surveys from two waves of data collected before and after participation in *Elevate Self Growth*. Several areas of well-being were measured pre- and post-program implementation using a *repeated measures* design. Baseline data was collected prior to participation in the *Elevate Self Growth* program (*n* = 15). After participation in *Elevate Self Growth* program, a follow-up survey was conducted (*n* = 11).

### 2.1. Baseline Survey

After enrolment in the *Elevate Self Growth* program, participants were assessed on several well-being areas (i.e., resilience, positive relationships, mindfulness, stress management, and purpose and meaning), participants’ knowledge, skill and environmental support related to well-being behaviors and performance of the specific well-being behaviors that underpinned the intervention (i.e., screen time reduction, quality leisure activities, physical activity, physical relaxation, meditation and improved sleep habits). Due to the small sample size, analysis was limited to descriptive statistics only. Comprehensive statistical tests were not performed due to the limited sample size and insufficient power. Online surveys were included in the first lesson of *Elevate Self Growth*.

The Brief Resilience Scale (BRS) was used to measure resilience [32]. While Windle et al. [33] deemed the 6-item BRS useful, to avoid survey fatigue, the 6-item BRS scale was considered, and 3 items were selected. Positive relationships were measured using the Positive Relationships subscale of the Psychological Well-Being questionnaire [34]. The Positive Relationships subscale has previously demonstrated reliability in assessing positive relationships [35]. Mindfulness was measured using a shortened MAAS scale [36]. MAAS was originally designed by Brown and Ryan [37], however, Osman et al. [36] deemed a 5-item MAAS useful. To stay in line with the survey structure, the 5-item scale was considered, and 3 items were selected. Stress was measured using the shortened Perceived Stress Scale [38], which has previously been validated by Vallejo et al. [39]. Purpose and meaning were measured with the presence subscale of the Meaning in Life Questionnaire [40]. Again, to stay in line with the survey structure and avoid survey fatigue, the 5-item scale was considered, and 3 items were selected. The scales were measured through the question “To what extent do you agree or disagree with the below statements?”. For each statement, respondents rated themselves using a 5-point Likert-type scale where 1 is ‘totally disagree’ and 5 is ‘totally agree’.

Knowledge, skill and environmental support were measured through direct statements related to each factor. For example, knowledge was measured through the statement “I know the benefits of (well-being behavior)”, skills were measured through the statement “I have the skills to (perform well-being behavior)”, and environment was measured through the statement “My environment enables me to (well-being behavior)”. Finally, the extent to which respondents were performing the well-being behaviors was measured by asking how many minutes they currently spent performing each behavior. A full list of questions can be found in Appendix A. Of the 19 people who enrolled in the *Elevate Self Growth* program, 15 completed the baseline survey, representing a response rate of 71%.

### 2.2. Follow-Up Survey

The follow-up survey was sent to participants who completed at least 25% of the proof-of-concept *Elevate Self Growth* program. Eleven people completed the follow-up survey, representing a response rate of 57% of all *Elevate Self Growth* participants.

### 2.3. In-Depth Interviews

In-depth interviews were conducted to better understand peoples’ unique experiences with the *Elevate Self Growth* program. People who completed the follow-up survey were invited to participate in a phone call to discuss their experiences and provide feedback to improve *Elevate Self Growth*. Interviewees were asked to share their opinions about what they liked, what they didn’t like and how they thought the program could be improved. The questions were asked according to a feedback grid [41]. Everyone who completed the follow-up survey was willing to complete an in-depth interview (*n* = 11).

## 3. Results

On average, participants in the sample completed 61% of the program material available to them online (see Table 1). Analysis indicates that three people completed the full program, two people almost completed the online program, and the remaining six people made some progress, but stagnated at some point.

### 3.1. Evaluation of Changes in Knowledge, Skill and Environmental Support

Figure 1 depicts the changes in knowledge, skills and environmental support related to the behaviors targeted with the intervention. Overall, the charts indicate that knowledge only changed slightly, whereas skills and environment improved more notably. Participants’ knowledge of the benefits of targeted well-being behaviors was already high and improved only moderately. Participants’ skills related to targeted well-being behaviors improved notably after engaging with the program, especially for physical relaxation and reducing screen time. Similarly, the extent to which participants’ environment was conducive to performing targeted well-being behaviors was considerably higher after engaging with the program, especially for reducing screen time, engaging in quality leisure activities and achieving high-quality sleep.

### 3.2. Evaluation of Well-Being Behavior Change

Figure 2 depicts the changes in well-being behavior before and after the program. Each chart depicts how much time individuals spent on each behavior, except for the ‘quality sleep’ chart, which displays how confident participants felt in maintaining healthy sleep habits. Overall, the charts indicate that participants’ well-being behavior improved post-program. After engaging with the program, participants spent fewer hours on their screens and more hours on quality leisure activities, physical activity, physical relaxation and somewhat more time meditating. In addition, participants believed their behavior around maintaining healthy sleep habits had improved. However, the change in time spent on meditation was not significant, perhaps due to the relatively low levels of completion of these lessons (i.e., content related to meditation was featured only in the last 40% of the course).

### 3.3. Evaluation of Well-Being Areas

Both surveys measured well-being areas, including resilience, positive relationships, mindfulness, stress management and purpose and meaning (see Figure 3). The repeated outcome measures used in the surveys indicated that all well-being areas improved between the baseline (2021) and follow-up survey (2022) waves.

Figure 4 depicts participants’ progress and changes reported pre- and post-program participation. Program progress was provided by the eLearning portal and measures the percentage of course material each participant completed. Change rates were measured as a percentage increase between the baseline and follow-up phases. Specifically, the following calculation was used to measure the proportional growth per participant: the sum of all baseline scores minus the sum of all follow-up scores divided by the sum of all baseline scores. The figure indicates that, on average, the more progress each participant made, the higher their proportional growth on well-being measures they reported. A Pearson correlation coefficient was computed to assess the relationship between program progress and program change rates. A significant and strong positive correlation was indicated between the two variables: r (9) = −0.82, *p* = 0.002, suggesting that progress within the program is associated with the rates of change observed in intended outcome variables. This relationship further indicates the importance of increasing program progress for participants. The individual results used to populate Figure 4 are reported in Table 1.

### 3.4. Evaluation of Participant Experiences

This section provides a summary of qualitative feedback from the well-being program’s participants in three categories: likes, dislikes and ideas and improvements.

*Likes.* First, participants were asked what they liked about the *Elevate Self Growth* well-being program. The most common aspect that people liked was the mix between science and practicality. Participants mentioned that the presentation of a mix of scientific evidence and real-life stories, followed by practical action points, enabled them to adopt the suggested principles. The scientific approach helped people to see the importance of the concepts, whereas the personal stories made the concepts relatable to one’s life. Some participants mentioned that the action points were concrete, practical and simple to apply.


*“I thought it was really cool to see that the content was based on scientific resources, as well as being taught by someone who had experienced everything himself.”*

*—28-year-old, Dutch male, working full time.*


The second most frequently mentioned aspect that people liked was the delivery format of the program. Participants commented on the combination of recorded videos and assignments in a workbook and how that enabled them to apply the ideas in their life. Specifically, participants liked the relatively short duration of the videos (between 2 and 15 min), which made it easy for them to consume. Some participants thought the videos communicated clear messages and found the use of captions and graphics engaging, which aided the learning process.


*“I liked the structure of the course and that it includes a video, and then an exercise to really implement the teachings.”*

*—28-year-old, Dutch male, working full time.*



*“I love how short the lessons are and that it’s broken up with the little clips. I really can’t think of any tips to better your delivery so far.”*

*—34-year-old, Australian female, self-employed.*


The third most frequently mentioned aspect that people liked was that the program enabled participants to be more present and intentional.


*“I catch myself when this device [smartphone] is distracting me. And I know how to alter it, so it serves me rather than I serve it.”*

*—30-year-old, Dutch female, unemployed*



*“I was able to spend less time online, and more time in the real world.”*

*—29-year-old, Australian male, self-employed.*


The fourth most frequently mentioned aspect that people liked was the use of practical tools, including the mobile application, the digital declutter, the full body relaxation audio track and the habit tracker.


*“I can study whenever, wherever I want”*

*—31-year-old, Spanish female, working full time [about the mobile app].*



*“The lessons helped me to manage my time and control my habits. I engaged in a digital declutter and already started to regain my focus.”*

*—28-year-old, Dutch male, working full time [about the digital declutter].*


Additional proof-of-concept program components that people liked included lessons related to meditation, sleep and building habits. Some believed the techniques were versatile and applicable to multiple facets of their life. Some participants mentioned the usefulness of having ‘lifetime’ access to the program, so they could return whenever they wanted. Finally, some people noticed a positive transformation in their life as a result of the program.


*“I could really see a transformation. I’ve become more compassionate and kinder towards others, and I try to be more nurturing and caring toward other people.”*

*—29-year-old, Finnish female, Student.*



*“Now I have a morning routine, which has helped me be regular in my meditation practice, which is one of the things I value most. This transformation has affected not just my morning but my whole day. I feel I have more energy than before. I get more done in my day and I feel more fulfilled in my life.”*

*—23-year-old, British female, working part time.*


*Dislikes.* When participants were asked what they disliked about the proof-of-concept program, 3 items emerged. First, two people mentioned that having lifetime access reduced the urgency to complete the program.


*“It would’ve been better if access to the program would be revoked after a while, so I’m more pushed to engage with the material.”*

*—33-year-old, Australian female, working part time.*


The proof-of-concept was self-paced to enable flexibility in the learning process. However, two people mentioned that the lack of a set timeframe may have prevented them from making regular progress.


*“Nobody chased me to complete the course and that’s why I probably didn’t feel the urge to continue at some point.”*

*—31-year-old, Spanish female, working full time.*


Third, one person wanted to be able to consume the content by listening (an audio program), so they could study while running or driving.

*Ideas and improvements.* When participants were asked what could be improved in the well-being program, 4 distinct topics emerged. First, some participants suggested a clear timeline to complete the course should be set, to prevent participants from dropping out. Second, participants recommended a ratio of online and offline, and pre-recorded and live sessions and resources. Due to COVID-19 restrictions, the pilot delivery was entirely online, and participants felt it would not have been as engaging as in-person events, which is consistent with preferences indicated in the co-creation phase. Therefore, participants recommend hosting sessions in-person where possible. One participant even recommended hosting a live event to bring all participants together to foster a sense of community among participants. Furthermore, most of the online resources were pre-recorded and did not allow instant two-way communication and interaction. One participant mentioned that having the ability to engage and communicate in real time could encourage people to ‘show up’ more frequently. Third, several participants suggested having regular one-on-one check-ins with the program facilitator to keep them more accoun and increase their completion rates. Fourth, one person would have liked to hear other participants’ stories to learn about their experiences, which is now possible following delivery of the proof-of-concept program.

## 4. Discussion

Improving the mental well-being of individuals—especially now and in the future—requires new, co-designed, theory-based health interventions. This paper detailed the initial implementation and an additional explanation of how a theoretically informed evaluation can be applied to consider if a proof-of-concept is achieving intended outcomes. The paper advances understanding of the potential utility of a theory-driven intervention implementation in three ways. First, this paper details how an evaluation can be performed following delivery of a proof-of-concept program to consider if intended outcomes may be achieved. This paper provides a brief overview to explain how SCT was applied within the proof-of-concept program and details how SCT core constructs can be applied in a repeated measures design to gain insights to inform project planning. Second, this study provides insights into the potential for SCT constructs to be used to change well-being behaviors. This study reports that progress in *Elevate Self Growth* may change well-being behavior. Further, participants provided feedback, delivering important insights that can be used to further refine the program. Finally, the proof-of-concept indicates the value of dynamic evaluation. Each of these contributions is discussed in turn.

### 4.1. Monitoring and Evaluating a Theory-Driven Social Marketing Intervention

This study reflects on the learnings following delivery of a proof-of-concept, theory-driven social marketing intervention that aims to improve young peoples’ well-being. The program was underpinned by SCT and sought to improve well-being by changing three theory-based constructs: knowledge, skills and environment. An ongoing debate on the role of theory [42,43,44] is evident. This study provides an exploration of SCT to determine if it offers the potential to improve well-being. Drawing from a small sample, our study suggests that increases in knowledge and skills and changes in levels of environmental support may lead to changes in well-being behavior (i.e., screen time reduction, quality leisure activities, physical activity, physical relaxation, meditation and improved sleep habits). The observed data patterns suggest that changing behaviors may contribute to well-being improvement (i.e., resilience, positive relationships, mindfulness, stress management and purpose and meaning). This study provides some preliminary support for theory-driven intervention design; however, future research delivering a randomized control trial and longitudinal evaluation design is needed to draw definitive conclusions that can validate these preliminary findings.

### 4.2. Assessing the Utility of Social Cognitive Theory in Social Marketing Interventions

This study reports the evaluation of a proof-of-concept that aimed to identify if a program designed to deliver components in accordance with three SCT constructs across 10 behaviors associated with well-being offers potential to improve well-being. This study demonstrates that when a participant’s knowledge about well-being behavior is already high, significant changes in knowledge may not be expected. Data insights indicate that when participants’ skills and environmental support related to a well-being behavior were lower at baseline, their scores did notably improve in those constructs following engagement with the program. It seems that many people already know *what* behaviors are conducive to better physical and mental health, yet they may not know *how* to integrate those behaviors into their lifestyle. For example, most participants seemed to know that excessive screen time was not conducive to their well-being, yet their skills to reduce screen time and their ability to influence their environment to reduce screen time were significantly lower prior to program participation. Ample evidence of this knowledge–behavior gap in intervention science exists [45,46], demonstrating that there is an inconsistency between what people know and their ability to act on that knowledge. Reflecting on the descriptive analysis in this study, people may already know that excessive screen time is distracting them, making them less productive and negatively impacting their mental health. However, they may not have the skills to replace screen time behavior with another behavior that is more beneficial for their well-being. Alternatively, they are not aware of the influence of their environment on their excessive screen time, and they may not understand how they can change their environment (i.e., their smartphone, computer, TV, etc.) to provide them with the support they need to actively reduce screen time. In summary, this study indicates the potential value of using SCT to influence participants’ knowledge, skills and surrounding environment in order to support them to change behaviors known to improve well-being. Participants shared mostly positive feedback about their experience with the intervention. Participants appreciated the tailored intervention components, and thus the theory-driven intervention design. This indicates that SCT could be a useful framework to design and plan interventions to change behavior for the greater social good.

### 4.3. Dynamic Monitoring and Agile Tweaking

Many program evaluations happen after interventions are completed [47,48,49]. The insights from such post-program evaluations are limited only to a next round of program implementation, which may never come if the results are not satisfactory. The need for more dynamic monitoring from the start enables social marketers to identify the most (cost-) effective methods to change behavior early on. Early insights into the effectiveness of intervention components on participants’ knowledge, skills, environmental support and behavioral outcomes enable social marketing professionals to assess what is working and what is not. Such early insights can be used to maximize monetary and time investment, thus ensuring that investments are placed behind the intervention components that deliver the strongest performance. Agile tweaking enables professionals to adjust programs or other market offerings more quickly and ensure that resources are used efficiently and effectively. If early insights identify effective activities, available resources can be redirected to save time and money and improve overall program effectiveness. Optimizing the effectiveness of intervention components during an intervention will likely improve the changes in knowledge, skills and behavior upon intervention completion.

### 4.4. Limitations and Future Research

This study is not without limitations. It only presents a descriptive analysis of 11 participants who engaged in a proof-of-concept well-being program. Few participants completed the full program, and some enrolled participants had not started the program. A longitudinal, controlled research design is needed with sufficient participants to deliver a sufficient sample size to permit statistical testing and determination of effect sizes. Descriptive assessment indicates the potential for this proof-of-concept program, and funding is needed to deliver a scaled test of the iterated program. Additional work is recommended to assess other behaviors that this intervention targeted. Specifically, we recommend undertaking a three-arm randomized controlled trial to compare the current program with other well-being programs that are not theory-driven, as well as a control condition. Then, analytical methods, including hidden Markov modelling, can be applied to identify factors driving change.

A series of recommendations were indicated in this study. Program participants who failed to complete the program sought supportive mechanisms through online or in-person contact. An iterated program that includes one or more recommended components can be tested in a field trial delivering a similar sample size to permit a cost–benefit assessment. If higher completion rates and potentially enhanced outcomes had been observed, the relative costs of delivering a form of in-person support could be warranted. Finally, this paper describes a theoretically informed program’s potential to enhance well-being. Future research testing the application of alternate theories in well-being is recommended, along with research that tests interventions informed by SCT in other health and environmental contexts.

## 5. Conclusions

This study presented an evaluation of a co-created, well-being proof-of-concept based on theory. This study has contributed to understanding success and failures in applying SCT to improve well-being. This research considered whether changes in knowledge, skills and environmental support occurred, and it considered if observed changes may offer potential to contribute to improvements in well-being for program participants. Program improvements were identified. This study delivered further evidence that theory may indeed offer potential to contribute to positive program outcomes. Social marketers are advised to link intervention components to theoretical constructs and include theoretical measures in their evaluation to clearly determine the effectiveness of theory-based intervention design.

## Figures and Tables

**Figure 1 ijerph-19-05248-f001:**
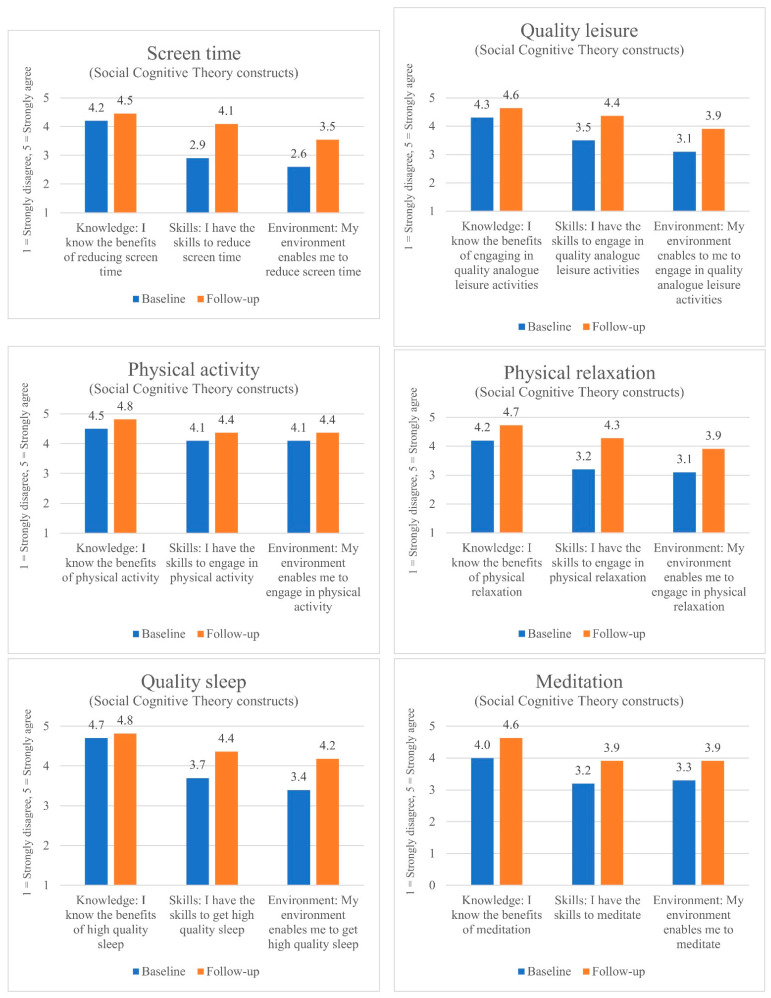
Changes in knowledge, skills and environment pre- and post-program.

**Figure 2 ijerph-19-05248-f002:**
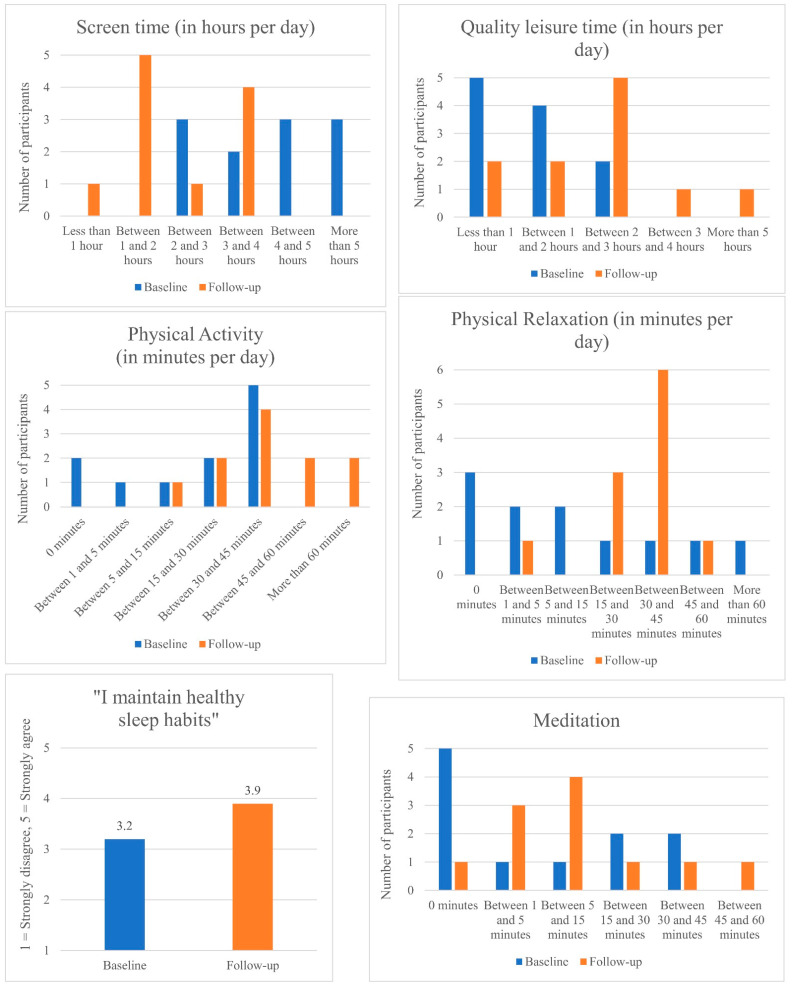
Changes in well-being behavior pre- and post-program.

**Figure 3 ijerph-19-05248-f003:**
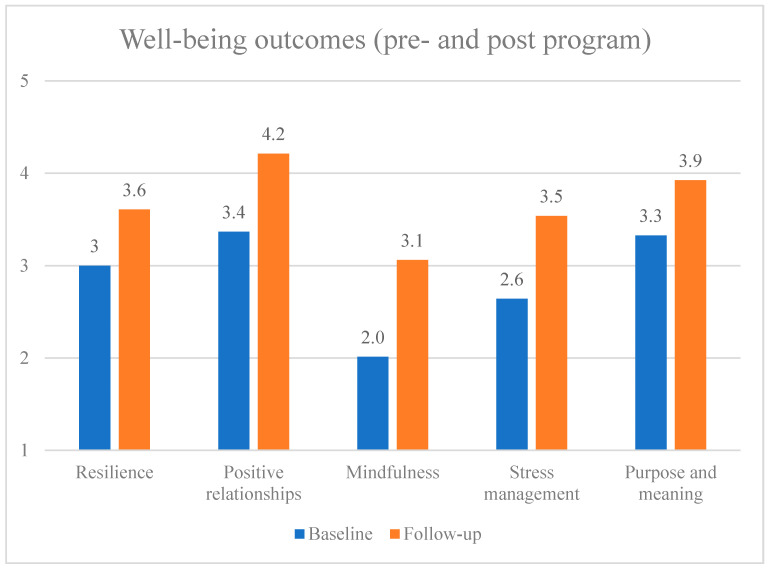
Evaluation of well-being areas (a higher number indicates a better score).

**Figure 4 ijerph-19-05248-f004:**
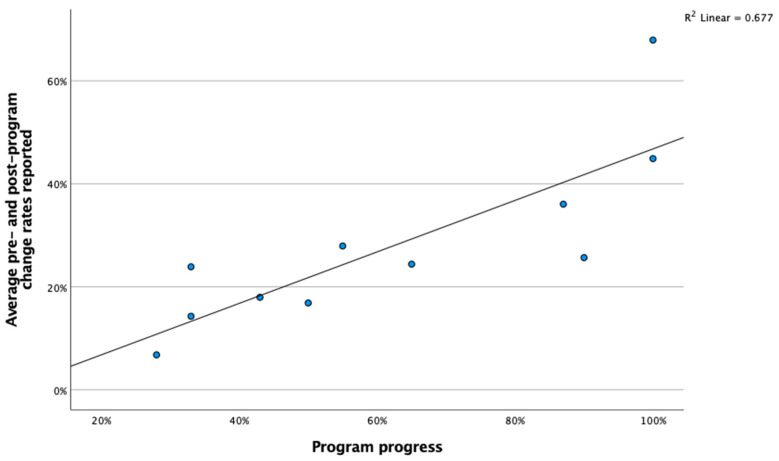
Well-being program progress versus reported change rates on well-being areas pre- and post-program (listed in percentages).

**Table 1 ijerph-19-05248-t001:** Evaluation of well-being areas (per individual).

	Program Progress	Average Pre- and Post-Program Change Rates Reported	Resilience		Positive Relationships		Mindfulness		Stress Management		Purpose and Meaning	
			Baseline	Follow-Up	Baseline	Follow-Up	Baseline	Follow-Up	Baseline	Follow-Up	Baseline	Follow-Up
23-year-old, British female, working part time	100%	68%	2.3	4.7	2.7	5.0	1.7	3.7	2.3	4.0	4.3	5.0
28-year-old, Dutch male, working full time	100%	45%	3.3	4.0	3.3	4.7	1.7	4.0	2.8	3.8	2.3	3.0
29-year-old, Finnish female, student	90%	26%	3.3	3.7	4.3	5.0	1.7	3.3	3.3	4.3	3.0	3.3
30-year-old, Dutch female, unemployed	87%	36%	3.3	4.0	3.7	5.0	2.0	3.0	1.7	3.0	4.0	5.0
31-year-old, Spanish female, working full time	65%	24%	1.7	2.7	3.3	4.0	2.0	3.0	3.0	3.8	4.0	4.0
34-year-old, Australian female, self-employed	55%	28%	3.7	3.3	4.3	4.7	1.0	2.3	2.3	3.8	3.7	5.0
24-year-old, Australian Male, working full-time	50%	17%	4.0	4.3	5.0	4.3	2.7	3.3	1.7	2.6	3.0	4.5
29-year-old, Australian male, self-employed	43%	18%	3.7	3.7	3.7	4.3	1.7	2.3	2.3	3.5	4.0	4.3
33-year-old, Australian female, working part time	33%	24%	3.3	4.0	1.3	3.3	3.0	3.0	4.0	4.3	3.3	4.0
42-year-old, Australian female, working full time	33%	14%	1.0	1.0	2.0	2.0	1.0	2.0	2.0	2.0	1.0	1.0
47-year-old, Dutch male, working full time	28%	7%	3.7	4.0	3.3	3.7	3.7	4.0	3.8	4.0	4.0	4.0
Mean values	61%	28%	3.0	3.6	3.4	4.2	2.0	3.1	2.6	3.5	3.3	3.9

## Data Availability

The data presented in this study are available on request from the corresponding author. The data are not publicly available to safeguard the privacy of program participants.

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
