# Peer review of "Improving Well-Being in Young Adults: A Social Marketing Proof-of-Concept"

_ijerph, 2022, doi:10.3390/ijerph19095248_

Round 1
Reviewer 1 Report
I want to thank the authors for this interesting contribution that outlines the development and initial evaluation of a well-being improvement program. Although I like the idea of the article, it still has some serious issues when it comes to its scientific quality. At this stage, it reads more like a promotional text than an academic publication and I hope that my suggestions can help the authors to improve upon that. For this purpose, I want to first summarize some general issues that should be tackled and then I will move on to issues more specifically related to the structure of the paper and the level of methodological details provided.
First, the motivation for this new well-being intervention should be strengthened. While it becomes clear that it is co-created, the theory connection throughout the paper is very weak. It should be made clear what other programs provide and how the current intervention improves upon them. In addition, as will be laid out in more detail below, how Social Cognitive Theory served as theoretical basis has to be made clear as it currently only seems to provide some means (i.e., variables) for the evaluation of the program. Further, the authors should be more careful with the claims that they make based on an exploratory evaluation of new intervention with a small sample size. For example, on p. 10 the authors argue that “This study provides an understanding of the effectiveness of SCT in changing behavior. It demonstrated that changing knowledge, skills and environmental support led to changes in well-being behavior. Consequently, this study indicated that these changes in well-being behavior (i.e., reducing screen time, and increasing physical activity, quality sleep, and meditation) led to improvements in several areas of well-being; namely positive relationships, mindfulness and stress.” If the authors really want to keep that claim, then I would demand a controlled, longitudinal experiment with control groups, a sufficiently large sample size and a comparison with alternative intervention programs in the next round of reviews.
Second, the structure of article also has to be reconsidered at some points. On p. 2, I recommend including a separate section that outlines the design of the intervention, starting from line 53. Here, the authors should also include the details on the program development that are presented on p. 3 (“Well-being program development”). Further, the authors should reconsider the value of the sections on p. 3-4 that mention in length how the program was promoted to potential participants (i.e., “Promotional Campaign”, “Well-being scorecard”, “The 7-Day Habit Challenge”, and “Well-being seminar”; also: What do the pictures of social media adverts in Figure 1 contribute to this article?). Instead, the focus here should be on the in-depth description of the actual intervention program. Some details become apart through comments in the qualitative data collection phase, but are not presented before.
Third, methodological details should be added at several points throughout the paper. On p. 2, it should be explained how “well-being” was defined for the participants of the participatory design study. Also on p. 2., the authors mention that they only focus on some aspects of well-being behaviors and well-being. Here it should also be stated whether the other were areas measured as well and are just not reported or whether the evaluation was only focusing on the presented aspects (and why on these aspects only). On p. 3 ff it should be made clear why Social Cognitive Theory was used as basis for this research, it should be briefly described in this context as well, and how it informed the development of the program. On p. 4, it is not clear how “knowledge”, “skill”, and “environmental support” link to the presented research at this point. It should be explained before this point that these are aspects of Social Cognitive Theory that are then directly connected to screen time, quality of sleep, physical activity, and meditation. On p. 5, it should be added what time lag the authors used between program completion and the second survey and what the motivation for the specific time lag was. On p. 7 – Figure 5: The authors should include an explanation for their calculation of “proportional growth”. In addition, I would highly recommend splitting up this chart into three (one for each dependent variable). Just mixing up some variables into one overall construct seems highly dubious without any further explanation. Here I would also suggest that the authors calculate basic correlations to add to the visual representation of the data. Finally, if the authors have trouble finding space for the detailed description of SCT and its value as well as the in-depth description of the program, I would suggest to simply reduce the extent of space that is filled with the comments from their qualitative evaluation on p. 8-10.
Finally, some minor issues that should be addressed as well:
Abstract: The last sentence seems out of place (lines 26-29)
- 6: Figure 2 – “Meditation determinants”, y-axis starts at 0
- 6: Figure 3 – “I maintain healthy sleep habits”: was the y-axis mislabeled here or why are there decimal places for baseline and follow-up?
- 8 – Table 1: Add “average” to “Pre- and Post-program change rates reported”
Overall, an interesting piece of research, though there is still much to be done before it is ready for publication.
Reviewer 2 Report
It is a very complete and exhaustive work. I would only recommend explaining the research objectives more concretely and relating them to the conclusions and discussion.
Reviewer 3 Report
A strength is that the program was developed based on a participatory approach involving 57 university students aged 18 to 35 years. Given the importance of this, it would be important for the authors to describe this in more detail. For example, how were students recruited, how were the sessions organized and conducted, how were data gathered and analyzed? Also, I wonder if they noticed any differences in the preferences of younger vs older students?
I would like to have a better understanding of the process to develop this theory-based intervention. What aspects of Social Cognitive Theory were used and how do these relate to the various aspects of the program. A graphic showing a logic model may be helpful here.
The program uses a social marketing approach. It would be helpful if this approach were defined and described in detail so the reader would have a clear understanding of how the program corresponds to social marketing principles and practices.
The evaluation is based on 11 participants and three of the six areas of the program. Some of these did not complete all aspects of the program. So, in my mind, this is insufficient to draw any conclusions about the potential worth of the program or the program development approach. I would suggest that the authors scale this “case study” evaluation back to a development piece, using the qualitative data to support their approach. Any kind of quantitative evaluation should be reserved for future research. In other words, this study could focus on ‘proof of concept’ rather than preliminary results.
Despite the fact that this is not a fulsome quantitative evaluation, no information was given on the survey items; how they were developed, by whom, how tested and validated, etc. If the authors focus on qualitative data, this would not be a problem for this paper.
It is not clear to me how the information on the promotional campaign fits into the program that is being reviewed. Were these means of attracting people to the program? How should the information that 560 people completed a well-being scorecard in 7 months or that 102 people attended one of the events be understood against the 11 that participated in completing questionnaire.
Round 2
Reviewer 1 Report
I want to thank the authors for their extensive revision and the effort that they put into this version of their manuscript. All of my comments have been sufficiently addressed and I consider the manuscript ready for publication.
Reviewer 3 Report
The authors have made significant changes to the manuscript which has resulted in a much improved presentation.